# The Convex Information Bottleneck Lagrangian

**DOI:** 10.3390/e22010098

**Published:** 2020-01-14

**Authors:** Borja Rodríguez Gálvez, Ragnar Thobaben, Mikael Skoglund

**Affiliations:** Department of Intelligent Systems, Division of Information Science and Engineering (ISE), KTH Royal Institute of Technology, 11428 Stockholm, Sweden

**Keywords:** information bottleneck, representation learning, mutual information, optimization

## Abstract

The information bottleneck (IB) problem tackles the issue of obtaining relevant compressed representations *T* of some random variable *X* for the task of predicting *Y*. It is defined as a constrained optimization problem that maximizes the information the representation has about the task, I(T;Y), while ensuring that a certain level of compression *r* is achieved (i.e., I(X;T)≤r). For practical reasons, the problem is usually solved by maximizing the IB Lagrangian (i.e., $L_{\mathrm{IB}}\left(T;\beta \right)=I\left(T;Y\right)-\beta I\left(X;T\right)$) for many values of β∈[0,1]. Then, the curve of maximal I(T;Y) for a given I(X;T) is drawn and a representation with the desired predictability and compression is selected. It is known when *Y* is a deterministic function of *X*, the IB curve cannot be explored and another Lagrangian has been proposed to tackle this problem: the squared IB Lagrangian: $L_{\mathrm{sq}-\mathrm{IB}}\left(T;\beta_{\mathrm{sq}}\right)=I\left(T;Y\right)-\beta_{\mathrm{sq}}I{\left(X;T\right)}^{2}$. In this paper, we (i) present a general family of Lagrangians which allow for the exploration of the IB curve in all scenarios; (ii) provide the exact one-to-one mapping between the Lagrange multiplier and the desired compression rate *r* for known IB curve shapes; and (iii) show we can approximately obtain a specific compression level with the convex IB Lagrangian for both known and unknown IB curve shapes. This eliminates the burden of solving the optimization problem for many values of the Lagrange multiplier. That is, we prove that we can solve the original constrained problem with a single optimization.

## 1. Introduction

Let X∈X and Y∈Y be two statistically dependent random variables with joint distribution $p_{\left(X,Y\right)}$. The information bottleneck (IB) [1] investigates the problem of extracting the relevant information from *X* for the task of predicting *Y*.

For this purpose, the IB defines a bottleneck variable T∈T obeying the Markov chain Y↔X↔T so that *T* acts as a representation of *X*. Tishby et al. [1] define the relevant information as the information the representation keeps from *Y* after the compression of *X* (i.e., I(T;Y)), provided a certain level of compression (i.e., I(X;T)≤r). Therefore, we select the representation which yields the value of the IB curve that best fits our requirements.

**Definition** **1** **(IB Functional).**
*Let X and Y be statistically dependent variables. Let *Δ* be the set of random variables T obeying the Markov condition Y↔X↔T. Then the IB functional is*
(1)$$F_{\mathrm{IB},\max}\left(r\right)=\max_{T\in \Delta }\left\{I(T;Y)\right\}\;s.t.\;I\left(X;T\right)\leq r,\;\forall r\in \left[0,\infty \right).$$


**Definition** **2** **(IB Curve).***The IB curve is the set of points defined by the solutions of*$F_{\mathrm{IB},\max}\left(r\right)$*for varying values of*r∈[0,∞).

**Definition** **3** **(Information Plane).**
*The plane is defined by the axes I(T;Y) and I(X;T).*


This method has been successfully applied to solve different problems from a variety of domains. For example:Supervised learning. In supervised learning, we are presented with a set of *n* pairs of input features and task outputs instances. We seek an approximation of the conditional probability distribution between the task outputs *Y* and the input features *X*. In classification tasks (i.e., when *Y* is a discrete random variable), the introduction of the variable *T* learned through the information bottleneck principle maintained the performance of standard algorithms based on the cross-entropy loss while providing with more adversarial attacks robustness and invariance to nuisances [2,3,4]. Moreover, by the nature of its definition the information bottleneck appears to be closely related with a trade-off between accuracy on the observable set and generalization to new, unseen instances (see Section 2).Clustering. In clustering, we are presented with a set of *n* pairs of instances of a random variable *X* and their attributes of interest *Y*. We seek groups of instances (or clusters *T*) such that the attributes of interest within the instances of each cluster are similar and the attributes of interest of the instances of different clusters are dissimilar. Therefore, the information bottleneck can be employed since it allows us to aim for attribute representative clusters (maximizing the similarity between instances within the clusters) and enforce a certain compression of the random variable *X* (ensuring a certain difference between instances of the different clusters). This has been successfully implemented, for instance, for gene expression analysis and word, document, stock pricing, or movie rating clustering [5,6,7].Image segmentation. In image segmentation, we want to partition an image into segments such that each pixel in a region shares some attributes. If we divide the image into very small regions *X* (e.g., each region is a pixel or a set of pixels defined by a grid), we can consider the problem of segmentation as that of clustering the regions *X* based on the region attributes *Y*. Hence, we can use the information bottleneck so that we seek region clusters *T* that are maximally informative about the attributes *Y* (e.g., the intensity histogram bins) and maintain a level of compression of the original regions *X* [8].Quantization. In quantization, we consider a random variable X∈X such that X is a large or continuous set. Our objective is to map *X* into a variable T∈T such that T is a smaller, countable set. If we fix the quantization set size to |T|=⌊r⌋ and aim at maximizing the information of the quantized variable with another random variable *Y* and restrict the mapping to be deterministic, then the problem is equivalent to the information bottleneck [9,10].Source coding. In source coding, we consider a data source S which generates a signal Y∈Y, which is later perturbed by a channel C:Y→X that outputs *X*. We seek a coding scheme that generates a code T∈T from the output of the channel *X* which is as informative as possible about the original source signal *Y* and can be transmitted at a small rate I(X;T)≤r. Therefore, this problem is equivalent to the the formulation of the information bottleneck [11].

Furthermore, it has been employed as a tool for development or explanation in other disciplines like reinforcement learning [12,13,14], attribution methods [15], natural language processing [16], linguistics [17] or neuroscience [18]. Moreover, it has connections with other problems such as source coding with side information (or the Wyner-Ahlswede-Körner (WAK) problem), the rate-distortion problem or the cost-capacity problem (see Sections 3, 6 and 7 from [19]).

In practice, solving a constrained optimization problem such as the IB functional is challenging. Thus, in order to avoid the non-linear constraints from the IB functional, the IB Lagrangian is defined.

**Definition** **4** **(IB Lagrangian).**
*Let X and Y be statistically dependent variables. Let *Δ* be the set of random variables T obeying the Markov condition Y↔X↔T. Then we define the IB Lagrangian as*
(2)$$L_{\mathrm{IB}}\left(T;\beta \right)=I\left(T;Y\right)-\beta I\left(X;T\right).$$


Here β∈[0,1] is the Lagrange multiplier which controls the trade-off between the information of *Y* retained and the compression of *X*. Note we consider β∈[0,1] because (i) for β≤0 many uncompressed solutions such as T=X maximize $L_{\mathrm{IB}}\left(T;\beta \right)$, and (ii) for β≥1 the IB Lagrangian is non-positive due to the data processing inequality (DPI) (Theorem 2.8.1 from Cover and Thomas [20]) and trivial solutions like T=const are maximizers with $L_{\mathrm{IB}}\left(T;\beta \right)=0$ [21].

We know the solutions of the IB Lagrangian optimization (if existent) are solutions of the IB functional by the Lagrange’s sufficiency theorem (Theorem 5 in Appendix A of Courcoubetis [22]). Moreover, since the IB functional is concave (Lemma 5 of Gilad-Bachrach et al. [19]) we know they exist (Theorem 6 in Appendix A of Courcoubetis [22]).

Therefore, the problem is usually solved by maximizing the IB Lagrangian with adaptations of the Blahut-Arimoto algorithm [1], deterministic annealing approaches [23] or a bottom-up greedy agglomerative clustering [6] or its improved sequential counterpart [24]. However, when provided with high-dimensional random variables *X* such as images, these algorithms do not scale well and deep learning-based techniques, where the IB Lagrangian is used as the objective function, prevailed [2,25,26].

Note the IB Lagrangian optimization yields a representation *T* with a given performance (I(X;T),I(T;Y)) for a given β. However, there is no one-to-one mapping between β and I(X;T). Hence, we cannot directly optimize for the desired compression level *r* but we need to perform several optimizations for different values of β and select the representation with the desired performance; e.g., [2]. The Lagrange multiplier selection is important since (i) sometimes even choices of β<1 lead to trivial representations such that $p_{T|X}=p_{T}$, and (ii) there exist some discontinuities on the performance level w.r.t. the values of β [27].

Moreover, recently Kolchinsky et al. [21] showed how in deterministic scenarios (such as many classification problems where an input $x_{i}$ belongs to a single particular class $y_{i}$) the IB Lagrangian could not explore the IB curve. Particularly, they showed that multiple β yielded the same performance level and that a single value of β could result in different performance levels. To solve this issue, they introduced the squared IB Lagrangian, $L_{\mathrm{sq}-\mathrm{IB}}\left(T;\beta_{\mathrm{sq}}\right)=I\left(T;Y\right)-\beta_{sq}I{\left(X;T\right)}^{2}$, which is able to explore the IB curve in any scenario by optimizing for different values of $\beta_{\mathrm{sq}}$. However, even though they realized a one-to-one mapping between $\beta_{\mathrm{sq}}$ and the compression level existed, they did not find such mapping. Hence, multiple optimizations of the Lagrangian were still required to find the best trade-off solution.

The main contributions of this article are:We introduce a general family of Lagrangians (the convex IB Lagrangians) which are able to explore the IB curve in any scenario for which the squared IB Lagrangian [21] is a particular case of. More importantly, the analysis made for deriving this family of Lagrangians can serve as inspiration for obtaining new Lagrangian families that solve other objective functions with intrinsic trade-offs such as the IB Lagrangian.We show that in deterministic scenarios (and other scenarios where the IB curve shape is known) one can use the convex IB Lagrangian to obtain a desired level of performance with a single optimization. That is, there is a one-to-one mapping between the Lagrange multiplier used for the optimization and the level of compression and informativeness obtained, and we provide the exact mapping. This eliminates the need for multiple optimizations to select a suitable representation.We introduce a particular case of the convex IB Lagrangians: the shifted exponential IB Lagrangian, which allows us to approximately obtain a specific compression level in any scenario. This way, we can approximately solve the initial constrained optimization problem from Equation (Equation 1) with a single optimization.

Furthermore, we provide some insight for explaining why there are discontinuities in the performance levels w.r.t. the values of the Lagrange multipliers. In a classification setting, we connect those discontinuities with the intrinsic clusterization of the representations when optimizing the IB bottleneck objective.

The structure of the article is the following: In Section 2 we motivate the usage of the IB in supervised learning settings. Then, in Section 3 we outline the important results used about the IB curve in deterministic scenarios. Later, in Section 4 we introduce the convex IB Lagrangian and explain some of its properties like the bijective mapping between Lagrange multipliers and the compression level and the range of such multipliers. After that, we support our (proved) claims with some empirical evidence on the MNIST [28] and TREC-6 [29] datasets in Section 5. Finally, in Section 6 we discuss our claims and empirical results. A PyTorch [30] implementation of the article can be found at https://github.com/burklight/convex-IB-Lagrangian-PyTorch.

In the Appendix A, Appendix B, Appendix C, Appendix D, Appendix E and Appendix F we provide with the proofs of the theoretical results. Then, in Appendix G we show some alternative families of Lagrangians with similar properties. Later, in Appendix H we provide with the precise experimental setup details to reproduce the results from the paper, and further experimentation with different datasets and neural network architectures. To conclude, in Appendix I we show some guidelines on how to set the convex information bottleneck Lagrangians for practical problems.

## 2. The IB in Supervised Learning

In this section, we will first give an overview of supervised learning in order to later motivate the usage of the information bottleneck in this setting.

### 2.1. Supervised Learning Overview

In supervised learning we are given a dataset $D_{n}={\left\{\left(x_{i},y_{i}\right)\right\}}_{i=1}^{n}$ of *n* pairs of input features and task outputs. In this case, *X* and *Y* are the random variables of the input features and the task outputs. We assume $x_{i}$ and $y_{i}$ are sampled i.i.d. from the true distribution $p_{\left(X,Y\right)}=p_{Y|X}p_{X}$. The usual aim of supervised learning is to use the dataset $D_{n}$ to learn a particular conditional distribution $q_{\hat{Y}|X}$ of the task outputs given the input features, parametrized by θ, which is a good approximation of $p_{Y|X}$. We use $\hat{Y}$ and $\hat{y}$ to indicate the predicted task output random variable and its outcome. We call a supervised learning task regression when *Y* is continuous-valued and classification when it is discrete.

Usually, supervised learning methods employ intermediate representations of the inputs before making predictions about the outputs; e.g., hidden layers in neural networks (Chapter 5 from Bishop [31]) or transformations in a feature space through the kernel trick in kernel machines like SVMs or RVMs (Sections 7.1 and 7.2 from Bishop [31]). Let *T* be a possibly stochastic function of the input features *X* with a parametrized conditional distribution $q_{T|X}$, then, *T* obeys the Markov condition Y↔X↔T. The mapping from the representation to the predicted task outputs is defined by the parametrized conditional distribution $q_{\hat{Y}|T}$. Therefore, in representation-based machine learning methods, the full Markov Chain is $Y↔X↔T↔\hat{Y}$. Hence, the overall estimation of the conditional probability $p_{Y|X}$ is given by the marginalization of the representations; i.e., $q_{\hat{Y}|X}=E_{t∼q_{T|X}}\left[q_{\hat{Y}|T=t}\right]$ (The notation $q_{\hat{Y}|T=t}$ represents the probability distribution $q_{\hat{Y}|T}\left(\cdot |t;\theta \right)$. For the rest of the text, we will use the same notation to represent conditional probability distributions where the conditioning argument is given).

In order to achieve the goal of having a good estimation of the conditional probability distribution $p_{Y|X}$, we usually define an instantaneous cost function 𝒿:X×Y→R. The value of this function 𝒿(x,y;θ) serves as a heuristic to measure the loss of our algorithm, parametrized by θ, obtains when trying to predict the realization of the task output *y* with the input realization *x*.

Clearly, we can be interested in minimizing the expectation of the instantaneous cost function over all the possible input features and task outputs, which we call the cost function. However, since we only have a finite dataset $D_{n}$ we have instead to minimize the empirical cost function.

**Definition** **5** **(Cost Function and Empirical Cost Function).**
*Let X and Y be the input features and task output random variables and x∈X and y∈Y their realizations. Let also 𝒿 be the instantaneous cost function, θ the parametrization of our learning algorithm, and $D_{n}={\left\{\left(x_{i},y_{i}\right)\right\}}_{i=1}^{n}$ the given dataset. Then, we define:*
(3)$$\begin{array}{cc} 1.\;The\;cost\;function: & J\left(p_{\left(X,Y\right)};\theta \right)=E_{\left(x,y\right)∼p_{\left(X,Y\right)}}\left[𝒿\left(x,y;\theta \right)\right] \end{array}$$
(4)$$\begin{array}{cc} 2.\;The\;emprical\;cost\;function: & \hat{J}\left(D_{n};\theta \right)=\frac{1}{n}\sum_{i=1}^{n}𝒿\left(x_{i},y_{i};\theta \right) \end{array}$$


The discrepancy between the normal and empirical cost functions is called the generalization gap or generalization error (see Section 1 of Xu and Raginsky [32], for instance) and intuitively, the smaller this gap is, the better our model generalizes; i.e., the better it will perform to new, unseen samples in terms of our cost function.

**Definition** **6** **(Generalization Gap).**
*Let $J(p_{\left(X,Y\right)};\theta )$ and $\hat{J}\left(D_{n};\theta \right)$ be the cost and the empirical cost functions as defined in Definition 5. Then, the generalization gap is defined as*
(5)$$\mathrm{gen}\left(D_{n};\theta \right)=J\left(p_{\left(X,Y\right)};\theta \right)-\hat{J}\left(D_{n};\theta \right),$$
*and it represents the error incurred when the selected distribution is the one parametrized by θ when the rule $\hat{J}\left(D_{n};\theta \right)$ is used instead of $J(p_{\left(X,Y\right)};\theta )$ as the function to minimize.*


Ideally, we would want to minimize the cost function. Hence, we usually try to minimize the empirical cost function and the generalization gap simultaneously. The modifications to our learning algorithm which intend to reduce the generalization gap but not hurt the performance on the empirical cost function are known as regularization.

### 2.2. Why Do We Use the IB?

**Definition** **7** **(Representation cross-entropy cost function).**
*Let X and Y be two statistically dependent variables with joint distribution $p_{\left(X,Y\right)}=p_{Y|X}p_{X}$. Let also T be a random variable obeying the Markov condition Y↔X↔T and $q_{T|X}$ and $q_{\hat{Y}|T}$ be the encoding and decoding distributions of our model, parametrized by θ. Finally, let $C(p_{Z}\left|\right|q_{Z})=-E_{z∼p_{Z}}\left[\log\left(q_{Z}\left(z\right)\right)\right]$ be the cross entropy between two probability distributions $p_{Z}$ and $q_{Z}$. Then, the cross-entropy cost function is*
(6)$$\begin{array}{c} J_{\mathrm{CE}}\left(p_{\left(X,Y\right)};\theta \right)=E_{\left(x,t\right)∼q_{T|X}p_{X}}\left[C(q_{Y|T=t}||q_{\hat{Y}|T=t})\right]=E_{\left(x,y\right)∼p_{\left(X,Y\right)}}\left[{𝒿}_{\mathrm{CE}}\left(x,y;\theta \right)\right], \end{array}$$
*where ${𝒿}_{\mathrm{CE}}\left(x,y;\theta \right)=-E_{t∼q_{T|X=x}}\left[q_{\hat{Y}|T=t}\left(y|t;\theta \right)\right]$ is the instantaneous representation cross-entropy cost function and $q_{Y|T}=E_{x∼p_{X}}\left[p_{Y|X=x}q_{T|X=x}/q_{T}\right]$ and $q_{T}=E_{x∼p_{X}}\left[q_{T|X=x}\right]$.*


The cross-entropy is a widely used cost function in classification tasks (e.g., Teahan [8], Krizhevsky et al. [33], Shore and Gray [34]) which has many interesting properties [35]. Moreover, it is known that minimizing the $J_{\mathrm{CE}}\left(p_{\left(X,Y\right)};\theta \right)$ maximizes the mutual information I(T;Y). That is:

**Proposition** **1** **(Minimizing the Cross Entropy Maximizes the Mutual Information).**
*Let $J_{\mathrm{CE}}\left(p_{\left(X,Y\right)};\theta \right)$ be the representation cross-entropy cost function as defined in Definition 7. Let also I(T;Y) be the mutual information between random variables T and Y in the setting from Definition 7. Then, minimizing $J_{\mathrm{CE}}\left(p_{\left(X,Y\right)};\theta \right)$ implies maximizing I(T;Y).*


The proof of this proposition can be found in Appendix A.

**Definition** **8** **(Nuisance).**
*A nuisance is any random variable that affects the observed data X but is not informative to the task we are trying to solve. That is, *Ξ* is a nuisance for Y if Y⊥Ξ or I(Ξ,Y)=0.*


Similarly, we know that minimizing I(X;T) minimizes the generalization gap for restricted classes when using the cross-entropy cost function (Theorem 1 of Vera et al. [36]), and when using I(T;Y) directly as an objective to maximize (Theorem 4 of Shamir et al. [37]). Furthermore, Achille and Soatto [38] in Proposition 3.1 upper bound the information of the input representations, *T*, with nuisances that affect the observed data, Ξ, with I(X;T). Therefore, minimizing I(X;T) helps generalization by not keeping useless information of Ξ in our representations.

Thus, jointly maximizing I(T;Y) and minimizing I(X;T) is a good choice both in terms of performance in the available dataset and in new, unseen data, which motivates studies on the IB.

## 3. The Information Bottleneck in Deterministic Scenarios

Kolchinsky et al. [21] showed that when *Y* is a deterministic function of *X* (i.e., Y=f(X)), the IB curve is piecewise linear. More precisely, it is shaped as stated in Proposition 2.

**Proposition** **2** **(The IB Curve is Piecewise Linear in Deterministic Scenarios).**
*Let X be a random variable and Y=f(X) be a deterministic function of X. Let also T be the bottleneck variable that solves the IB functional. Then the IB curve in the information plane is defined by the following equation:*
(7)$$\left\{\begin{array}{ccc} I(T;Y)=I(X;T) & \mathrm{if} & I(X;T)\in [0,I(X;Y))\\ I(T;Y)=I(X;Y) & \mathrm{if} & I(X;T)\geq I(X;Y) \end{array}\right.$$


Furthermore, they showed that the IB curve could not be explored by optimizing the IB Lagrangian for multiple β because the curve was not strictly concave. That is, there was not a one-to-one relationship between β and the performance level.

**Theorem** **1** **(In Deterministic Scenarios, the IB Curve cannot be Explored Using the IB Lagrangian).**
*Let X be a random variable and Y=f(X) be a deterministic function of X. Let also *Δ* be the set of random variables T obeying the Markov condition Y↔X↔T. Then:*
*1.* 
*Any solution T∈Δ such that I(X;T)∈[0,I(X;Y)) and I(T;Y)=I(X;T) solves ${arg max}_{T\in \Delta }\left\{L_{\mathrm{IB}}\left(T;\beta \right)\right\}$ for β=1. That is, many different compression and performance levels can be achieved for β=1.*
*2.* 
*Any solution T∈Δ such that I(X;T)>I(X;Y) and I(T;Y)=I(X;Y) solves ${arg sup}_{T\in \Delta }\left\{L_{\mathrm{IB}}\left(T;\beta \right)\right\}$ for β=0. That is, many compression levels can be achieved with the same performance for β=0.*

*Note we use the supremum in this case since for β=0 we have that I(X;T) could be infinite and then the search set from Equation (Equation 1); i.e., {T:Y↔X↔T}∩{T:I(X;T)<∞} is not compact anymore.*
*3.* *Any solution*T∈Δ*such that*I(X;T)=I(T;Y)=I(X;Y)*solves*${arg max}_{T\in \Delta }\left\{L_{\mathrm{IB}}\left(T;\beta \right)\right\}$*for all*β∈(0,1). *That is, many different β achieve the same compression and performance level.*


An alternative proof for this theorem can be found in Appendix B.

## 4. The Convex IB Lagrangian

### 4.1. Exploring the IB Curve

Clearly, a situation like the one depicted in Theorem 1 is not desirable, since we cannot aim for different levels of compression or performance. For this reason, we generalize the effort from Kolchinsky et al. [21] and look for families of Lagrangians which are able to explore the IB curve. Inspired by the squared IB Lagrangian, $L_{\mathrm{sq}-\mathrm{IB}}\left(T;\beta_{\mathrm{sq}}\right)=I\left(T;Y\right)-\beta_{\mathrm{sq}}I{\left(X;T\right)}^{2}$, we look at the conditions a function of I(X;T) requires in order to be able to explore the IB curve. In this way, we realize that any monotonically increasing and strictly convex function will be able to do so, and we call the family of Lagrangians with these characteristics the convex IB Lagrangians, due to the nature of the introduced function.

**Theorem** **2** **(Convex IB Lagrangians).**
*Let *Δ* be the set of r.v. T obeying the Markov condition Y↔X↔T. Then, if u is a monotonically increasing and strictly convex function, the IB curve can always be recovered by the solutions of ${arg max}_{T\in \Delta }\left\{L_{\mathrm{IB},u}\left(T;\beta_{u}\right)\right\}$, with*
(8)$$L_{\mathrm{IB},u}\left(T;\beta_{u}\right)=I\left(T;Y\right)-\beta_{u}u\left(I\left(X;T\right)\right).$$

*That is, for each point (I(X;T),I(T;Y)) s.t. dI(T;Y)/dI(X;T)>0 there is a unique $\beta_{u}$ for which maximizing $L_{\mathrm{IB},u}\left(T;\beta_{u}\right)$ achieves this solution. Furthermore, $\beta_{u}$ is strictly decreasing w.r.t. I(X;T). We call $L_{\mathrm{IB},u}\left(T;\beta_{u}\right)$ the convex IB Lagrangian.*


The proof of this theorem can be found in Appendix C. Furthermore, by exploiting the IB curve duality (Lemma 10 of Gilad-Bachrach et al. [19]) we were able to derive other families of Lagrangians which allow for the exploration of the IB curve (Appendix G).

**Remark** **1.**
*Clearly, we can see how if u is the identity function (i.e., u(I(X;T))=I(X;T)) then we end up with the normal IB Lagrangian. However, since the identity function is not strictly convex, it cannot ensure the exploration of the IB curve.*


During the proof of this theorem we observed a relationship between the Lagrange multipliers and the solutions obtained of the normal IB Lagrangian $L_{\mathrm{IB}}\left(T;\beta \right)$ and the convex IB Lagrangian $L_{\mathrm{IB},u}\left(T;\beta_{u}\right)$. This relationship is formalized in the following corollary.

**Corollary** **1** **(IB Lagrangian and IB convex Lagrangian connection).**
*Let $L_{\mathrm{IB}}\left(T;\beta \right)$ be the IB Lagrangian and $L_{\mathrm{IB},u}\left(T;\beta_{u}\right)$ the convex IB Lagrangian. Then, maximizing $L_{\mathrm{IB}}\left(T;\beta \right)$ and $L_{\mathrm{IB},u}\left(T;\beta_{u}\right)$ can obtain the same point in the IB curve if $\beta_{u}=\beta /u^{\prime }\left(I\left(X;T\right)\right)$, where $u^{\prime }$ is the derivative of u.*


This corollary allows us to better understand why the addition of *u* allows for the exploration of the IB curve in deterministic scenarios. If we note that for β=1 we can obtain any point in the increasing region of the curve, then we clearly see how evaluating $u^{\prime }$ for different values of I(X;T) define different values of $\beta_{u}$ that obtain such points. Moreover, it lets us see how if for β=0 maximizing the IB Lagrangian could obtain any point (I(X;Y);I(X;T)) with I(X;T)>I(X;Y), then the same happens for the IB convex Lagrangian.

### 4.2. Aiming for a Specific Compression Level

Let $B_{u}$ denote the domain of Lagrange multipliers $\beta_{u}$ for which we can find solutions in the IB curve with the convex IB Lagrangian. Then, the convex IB Lagrangians do not only allow us to explore the IB curve with different $\beta_{u}$. They also allow us to identify the specific $\beta_{u}$ that obtains a given point (I(X;T),I(T;Y)), provided we know the IB curve in the information plane. Conversely, the convex IB Lagrangian allows finding the specific point (I(X;T),I(T;Y)) that is obtained by a given $\beta_{u}$.

**Proposition** **3** **(Bijective Mapping between IB Curve Point and Convex IB Lagrange multiplier).**
*Let the IB curve in the information plane be known; i.e., $I\left(T;Y\right)=f_{\mathrm{IB}}\left(I\left(X;T\right)\right)$ is known. Then there is a bijective mapping from Lagrange multipliers $\beta_{u}\in B_{u}\backslash \left\{0\right\}$ from the convex IB Lagrangian to points in the IB curve $(I\left(X;T\right),f_{\mathrm{IB}}\left(I\left(X;T\right)\right)$. Furthermore, these mappings are:*
(9)$$\beta_{u}=\frac{df_{\mathrm{IB}}\left(I\left(X;T\right)\right)}{dI(X;T)}\frac{1}{u^{\prime }\left(I\left(X;T\right)\right)}\;and\;I\left(X;T\right)={\left(u^{\prime }\right)}^{-1}\left(\frac{df_{\mathrm{IB}}\left(I\left(X;T\right)\right)}{dI(X;T)}\frac{1}{\beta_{u}}\right),$$
*where $u^{\prime }$ is the derivative of u and ${\left(u^{\prime }\right)}^{-1}$ is the inverse of $u^{\prime }$.*


This is especially interesting since in deterministic scenarios we know the shape of the IB curve (Theorem 2) and since the convex IB Lagrangians allow for the exploration of the IB curve (Theorem 2). A proof for Proposition 3 can be found in Appendix D.

**Remark** **2.**
*Note that the definition from Tishby et al. [1] $\beta =df_{\mathrm{IB}}\left(I\left(X;T\right)\right)/dI\left(X;T\right)$ only allows for a bijection between β and I(X;T) if $f_{\mathrm{IB}}$ is a strictly convex, and known function, and we have seen this is not the case in deterministic scenarios (Theorem 1).*


A direct result derived from this proposition is that we know the domain of Lagrange multipliers, $B_{u}$, which allows for the exploration of the IB curve if the shape of the IB curve is known. Furthermore, if the shape is not known we can at least bound that range.

**Corollary** **2** **(Domain of Convex IB Lagrange Multiplier with Known IB Curve Shape).**
*Let the IB curve in the information plane be $I\left(T;Y\right)=f_{\mathrm{IB}}\left(I\left(X;T\right)\right)$ and let $I_{\max}=I\left(X;Y\right)$. Let also $I\left(X;T\right)=r_{\max}$ be the minimum mutual information s.t. $f_{\mathrm{IB}}\left(r_{\max}\right)=I_{\max}$; i.e., $r_{\max}={arg inf}_{r}\left\{f_{\mathrm{IB}}\left(r\right)\right\}s.t.f_{\mathrm{IB}}\left(r\right)=I_{\max}$. Then, the range of Lagrange multipliers that allow the exploration of the IB curve with the convex IB Lagrangian is $B_{u}=\left[\beta_{u,\min},\beta_{u,\max}\right]$, with*
(10)$$\beta_{u,\min}=\lim_{r\to r_{\max}^{-}}\left\{\frac{f_{\mathrm{IB}}^{\prime }\left(r\right)}{u^{\prime }\left(r\right)}\right\}\;and\;\beta_{u,\max}=\lim_{r\to 0^{+}}\left\{\frac{f_{\mathrm{IB}}^{\prime }\left(r\right)}{u^{\prime }\left(r\right)}\right\},$$
*where $f_{\mathrm{IB}}^{\prime }\left(r\right)$ and $u^{\prime }\left(r\right)$ are the derivatives of $f_{\mathrm{IB}}\left(I\left(X;T\right)\right)$ and u(I(X;T)) w.r.t. I(X;T) evaluated at r respectively. Also, note that there are some scenarios where $r_{\max}\to \infty$ (see, e.g., [39]), in these scenarios $\beta_{u,\min}=\lim_{r\to \infty }\left\{f_{\mathrm{IB}}^{\prime }\left(r\right)/u^{\prime }\left(r\right)\right\}\geq 0$.*


**Corollary** **3** **(Domain of Convex IB Lagrange Multiplier Bound).**
*The range of the Lagrange multipliers that allow the exploration of the IB curve is contained by $\left[0,\beta_{u,\mathrm{top}}\right]$ which is also contained by $\left[0,\beta_{u,\mathrm{top}}^{+}\right]$, where*
(11)$$\beta_{u,\mathrm{top}}=\frac{{\left(\inf_{\Omega_{x}\subset X}\left\{\beta_{0}\left(\Omega_{x}\right)\right\}\right)}^{-1}}{\lim_{r\to 0^{+}}\left\{u^{\prime }\left(r\right)\right\}},\;and\;\beta_{u,\mathrm{top}}^{+}=\frac{1}{\lim_{r\to 0^{+}}\left\{u^{\prime }\left(r\right)\right\}},$$
*where $u^{\prime }\left(r\right)$ is the derivative of u(I(X;T)) w.r.t. I(X;T) evaluated at r, X is the set of possible realizations of X and $\beta_{0}$ and $\Omega_{x}$ are defined as in [27] (Note in [27] they consider the dual problem (see Appendix G), so when they refer to $\beta^{-1}$ it translates to β in this article). That is, $B_{u}\subseteq \left[0,\beta_{u,\mathrm{top}}\right]\subseteq \left[0,\beta_{u,\mathrm{top}}^{+}\right]$.*


Corollaries 2 and 3 allow us to reduce the range search for β when we want to explore the IB curve. Practically, $\inf_{\Omega_{x}\subset X}\left\{\beta_{0}\left(\Omega_{x}\right)\right\}$ might be difficult to calculate so Wu et al. [27] derived an algorithm to approximate it. However, we still recommend setting the numerator to 1 for simplicity. The proofs for both corollaries are found in Appendix E and Appendix F.

## 5. Experimental Support

In order to showcase our claims we use the MNIST [28] and the TREC-6 [29] datasets. We modify the nonlinear-IB method [26], which is a neural network that minimizes the cross-entropy while also minimizing a differentiable kernel-based estimate of I(X;T) [40]. Then, we used this technique to maximize a lower bound on the convex IB Lagrangians by applying the functions *u* to the I(X;T) estimate.

The network structure is the following: first, a stochastic encoder $T=f_{\mathrm{enc}}\left(X;\theta \right)+W$ with $p_{W}=N\left(0,I_{d}\right)$ such that $T\in R^{d}$, where *d* is the dimension of the bottleneck variable (Note that the encoder needs to be stochastic to (i) ensure a finite and well-defined mutual information [21,41] and (ii) make gradient-based optimization methods over the IB Lagrangian useful [41]). Second, a deterministic decoder $q_{\hat{Y}|T}=f_{\mathrm{dec}}\left(T;\theta \right)$. For the MNIST dataset both the encoder and the decoder are fully-connected networks, for a fair comparison with [26]. For the TREC-6 dataset, the encoder is a set of convolutions of word embeddings followed by a fully-connected network and the decoder is also a fully-connected network. For further details about the experiment setup, additional results for different values of α and η and supplementary experimental results for different datasets and network architectures, please refer to Appendix H.

In Figure 1 we show our results for two particularizations of the convex IB Lagrangians:the power IB Lagrangians: $L_{\mathrm{IB},\mathrm{pow}}\left(T;\beta_{\mathrm{pow}},\alpha \right)=I\left(T;Y\right)-\beta_{\mathrm{pow}}I{\left(X;T\right)}^{\left(1+\alpha \right)}$, α>0 (Note when α=1 we have the squared IB functional from Kolchinsky et al. [21]).the exponential IB Lagrangians: $L_{\mathrm{IB},\exp}\left(T;\beta_{\exp},\eta \right)=I\left(T;Y\right)-\beta_{\exp}\exp\left(\eta I\left(X;T\right)\right)$, η>0.

We can clearly see how both Lagrangians are able to explore the IB curve (first column from Figure 1) and how the theoretical performance trend of the Lagrangians matches the experimental results (second and third columns from Figure 1). There are small mismatches between the theoretical and experimental performance. This is because using the nonlinear-IB, as stated by Kolchinsky et al. [21], does not guarantee that we find optimal representations due to factors like (i) inaccurate estimation of I(X;T), (ii) restrictions on the structure of *T*, (iii) use of an estimation of the decoder instead of the real one and (iv) the typical non-convex optimization issues that arise with gradient-based methods. The main difference comes from the discontinuities in performance for increasing β, which cause is still unknown (cf. Wu et al. [27]). It has been observed, however, that the bottleneck variable performs an intrinsic clusterization in classification tasks (see, for instance, [21,26,42] or Figure 2b). We observed how this clusterization matches with the quantized performance levels observed (e.g., compare Figure 2a with the top center graph in Figure 1); with maximum performance when the number of clusters is equal to the cardinality of *Y* and reducing performance with a reduction of the number of clusters, which is in line with the concurrent work from Wu and Fischer [43]. We do not have a mathematical proof for the exact relationship between these two phenomena; however, we agree with Wu et al. [27] that it is an interesting matter and hope this observation serves as motivation to derive new theory.

In practice, there are different criteria for choosing the function *u*. For instance, the exponential IB Lagrangian could be more desirable than the power IB Lagrangian when we want to draw the IB curve since it has a finite range of $\beta_{u}$. This is $B_{u}=\left[{\left(\eta \exp\left(\eta I_{\max}\right)\right)}^{-1},\eta^{-1}\right]$ for the exponential IB Lagrangian vs. $B_{u}=\left[{\left(\left(1+\alpha \right)I_{\max}^{\alpha }\right)}^{-1},\infty \right)$ for the power IB Lagrangian. Furthermore, there is a trade-off between (i) how much the selected *u* function resembles a linear function in our region of interest; e.g., with α or η close to zero, since it will suffer from similar problems as the original IB Lagrangian; and (ii) how fast it grows in our region of interest; e.g., higher values of α or η, since it will suffer from value convergence; i.e., optimizing for separate values of $\beta_{u}$ will achieve similar levels of performance (Figure 3). Please, refer to Appendix I for a more thorough explanation of these two phenomena.

Particularly, the value convergence phenomenon can be exploited in order to approximately obtain a particular level of compression $r^{*}$, both for known and unkown IB curves (see Appendix I or the example in Figure 4). For known IB curves, we also know the achieved predictability I(T;Y) since it is the same as the level of compression I(X;T). For this exploitation, we can employ the shifted version of the exponential IB Lagrangian (which is also a particular case of the convex IB Lagrangian):the shifted exponential IB Lagrangians:
$$L_{\mathrm{IB},\mathrm{sh}-\exp}\left(T;\beta_{\mathrm{sh}-\exp},\eta ,r^{*}\right)=I\left(T;Y\right)-\beta_{\mathrm{sh}-\exp}\exp\left(\eta \left(I\left(X;T\right)-r^{*}\right)\right),\eta >0,r^{*}\in \left[0,\infty \right).$$

For this Lagrangian, the optimization procedure converges to representations with approximately the desired compression level $r^{*}$ if the hyperparameter η is set to a large value.

In Figure 4 we show the results of aiming for a compression level of $r^{*}=2$ bits in the MNIST dataset and of $r^{*}=16$ bits in the TREC-6 dataset, both with η=200. We can see how for different values of $\beta_{\mathrm{sh}-\exp}$ we can obtain the same desired compression level, which makes this method stable to variations in the Lagrange multiplier selection.

To sum up, in order to achieve a desired level of performance with the convex IB Lagrangian as an objective one should:In a deterministic or close to a deterministic setting (see ϵ-deterministic definition in Kolchinsky et al. [21]): Use the adequate $\beta_{u}$ for that performance using Proposition 3. Then if the performance is lower than desired, i.e., we are placed in the wrong performance plateau, gradually reduce the value of $\beta_{u}$ until reaching the previous performance plateau. Alternatively, exploit the value convergence phenomenon with, for instance, the shifted exponential IB Lagrangian.In a stochastic setting: exploit the value convergence phenomenon with, for instance, the shifted exponential IB Lagrangian. Alternatively, draw the IB curve with multiple values of $\beta_{u}$ on the range defined by Corollary 3 and select the representations that best fit their interests.

## 6. Conclusions

The information bottleneck is a widely used and studied technique. However, it is known that the IB Lagrangian cannot be used to achieve varying levels of performance in deterministic scenarios. Moreover, in order to achieve a particular level of performance, multiple optimizations with different Lagrange multipliers must be done to draw the IB curve and select the best traded-off representation.

In this article we introduced a general family of Lagrangians which allow to (i) achieve varying levels of performance in any scenario, and (ii) pinpoint a specific Lagrange multiplier $\beta_{u}$ to optimize for a specific performance level in known IB curve scenarios; e.g., deterministic. Furthermore, we showed the $\beta_{u}$ domain when the IB curve is known and a $\beta_{u}$ domain bound for exploring the IB curve when it is unknown. This way we can reduce and/or avoid multiple optimizations and, hence, reduce the computational effort for finding well traded-off representations. Moreover, (iii) when the IB curve is not known, we saw how we can exploit the value convergence issue of the convex IB Lagrangian to approximately obtain a specific compression level for both known and unknown IB curve shapes. Finally, (iv) we provided some insight into the discontinuities on the performance levels w.r.t. the Lagrange multipliers by connecting those with the intrinsic clusterization of the bottleneck variable.

## Figures and Tables

**Figure 1 entropy-22-00098-f001:**
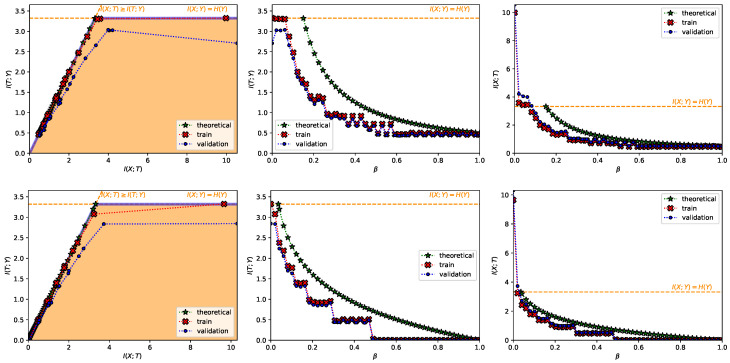
The top row shows the results for the power information bottleneck (IB) Lagrangian with α=1, and the bottom row for the exponential IB Lagrangian with η=1, both in the MNIST dataset. In each row, from left to right it is shown (i) the information plane, where the region of possible solutions of the IB problem is shadowed in light orange and the information-theoretic limits are the dashed orange line; (ii) I(T;Y) as a function of $\beta_{u}$; and (iii) the compression I(X;T) as a function of $\beta_{u}$. In all plots, the red crosses joined by a dotted line represent the values computed with the training set, the blue dots the values computed with the validation set and the green stars the theoretical values computed as dictated by Proposition 3. Moreover, in all plots, it is indicated $I\left(X;Y\right)=H\left(Y\right)=\log_{2}\left(10\right)$ in a dashed, orange line. All values are shown in bits.

**Figure 2 entropy-22-00098-f002:**
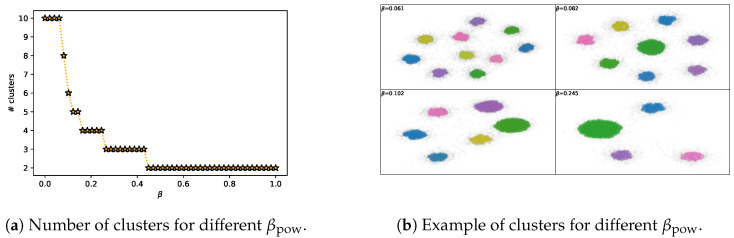
Depiction of the clusterization behavior of the bottleneck variable for the power IB Lagrangian in the MNIST dataset with α=1. The clusters were obtained using the DBSCAN algorithm [44,45].

**Figure 3 entropy-22-00098-f003:**
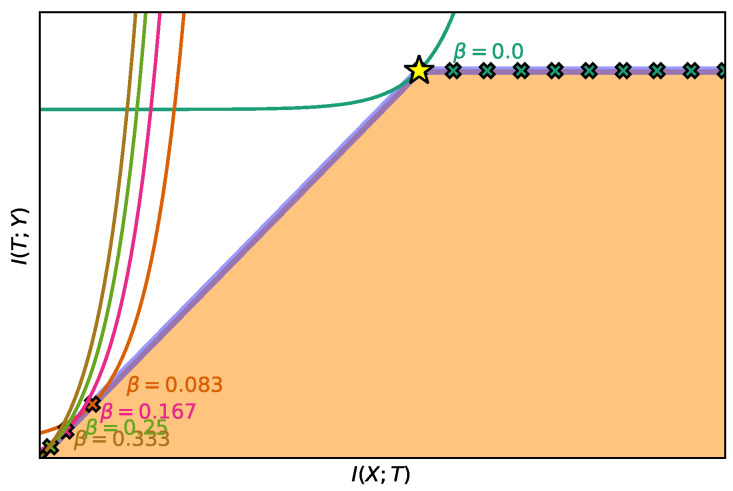
Example of value convergence with the exponential IB Lagrangian with η=3. We show the intersection of the isolines of $L_{\mathrm{IB},\exp}\left(T;\beta_{\exp}\right)$ for different $\beta_{\exp}\in B_{\exp}\approx \left[1.56\times 10^{-5},3^{-1}\right]$ using Corollary 2.

**Figure 4 entropy-22-00098-f004:**
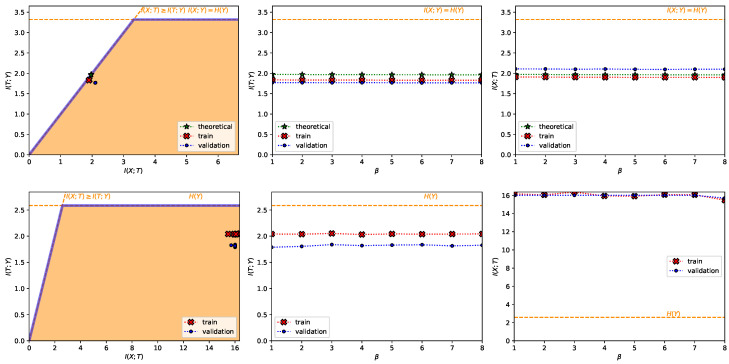
Example of value convergence exploitation with the shifted exponential Lagrangian with η=200. In the top row, for the MNIST dataset aiming for a compression level $r^{*}=2$ and in the bottom row, for the TREC-6 dataset aiming for a compression level of $r^{*}=16$. In each row, from left to right it is shown (i) the information plane, where the region of possible solutions of the IB problem is shadowed in light orange and the information-theoretic limits are the dashed orange line; (ii) I(T;Y) as a function of $\beta_{u}$; and (iii) the compression I(X;T) as a function of $\beta_{u}$. In all plots, the red crosses joined by a dotted line represent the values computed with the training set, the blue dots the values computed with the validation set and the green stars the theoretical values computed as dictated by Proposition 3. Moreover, in all plots, it is indicated H(Y) in a dashed, orange line. All values are shown in bits.

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
