# Peer review of "The Convex Information Bottleneck Lagrangian"

_entropy, 2020, doi:10.3390/e22010098_

Round 1

Reviewer 1 Report

This paper introduces a family of convex Information Bottleneck (IB) Lagrangian, which the authors proved can explore full IB curve, which is useful for deterministic or near-deterministic scenarios. The authors further derived the one-to-one correspondence between the Lagrange multiplier \beta_u and I(X;T), the desired level of compression.

For the technical point of view, this paper is relatively incremental. On the other hand, I think that the low originality may be compensated by a more thorough experimental study and results. For example:

In practical scenarios, how to choose the u function? The author provides some experiments and guidelines, but it would be nicer if more concrete guidelines are provided, and also how the increased level of stochasticity (thus increase level of strong concavity for the IB curve) may affect the choice of u.

Some more interesting experiment study may be:

Is it possible to design the convex IB Lagrangian to target a specific region? In the case where the IB curve is unknown, how would you choose u? For example, perform some exploration first to estimate a rough shape, then design u, and iterate?

Some concrete result like the above will make the paper much more useful and interesting from a practical point of view.

Some technical details:

For Corollary 2, the authors assume that there exists minimum I(X; T)=r such tha f_IB(r) = I_max. However, there are scenarios where the r required is infinity (see the paper "Strong Data Processing Inequalities for Input Constrained Additive Noise Channels" by du Pin Calmon et al. The author should clarify that r may be infinity.

Some typos:

Fig.1 and Fig.3 For the middle column, the y label should be I(T;Y) Line 395 and 396: "decreasing" should be "increasing"

Author Response

Dear Reviewer,

First of all, thank you very much for your comments and suggestions. We believe the manuscript has improved after we revised it with them in mind.

All the revisions of the manuscript (including typos) are included in red as suggested by the editor. The additions based on your comments are:

More experimentation with other datasets and network architectures. The exploitation of value convergence of the convex IB Lagrangians in order to approximately obtain any compression level for both known and unknown IB curve shapes. This has been motivated by your comment on targeting a specific region. We believe this point covers some of your suggestions for improvement:
- How to choose the u function in practical scenarios.
- How to target a specific region. In particular we target a specific compression level.
- How to choose the u function when the IB curve is unknown. We recommend the exploitation of the value convergence phenomenon. A note (note 3) on the possibility of r growing to infinity and the substitution of 'minimum' to 'infimum' in Corollary 2.

Other than that, we added some more context in the introduction of the application of the IB in different domains.

Sincerely,

Reviewer 2 Report

In this manuscript, the authors propose a new family of Lagrangians for the Information Bottleneck (IB) problem. It is known that using the IB Lagrangian is problematic in case of deterministic scenarios, as the desired level of performance cannot be achieved with a single optimization, and multiple optimizations are needed for this purpose. Here, this is avoided by a convex Lagrangian and the proposed approach seems to be an ideal one to take as a reference in deriving other Lagrangians for different loss functions.

The background is sufficiently provided by the authors with a brief overview of the IB problem definition, IB curves and the optimization mentality. The introduction involves the relevant literature survey and the text contains satisfactory comparisons. The text is somewhat easy to follow smoothly, once the references are followed accordingly. The method is explained clearly by mathematical proofs and computer simulations.

Therefore, I think this paper will be a good contribution to the Entropy journal and the relevant scientific society.

Author Response

Dear Reviewer,

First of all, thank you very much for your comments.

All the revisions of the manuscript (including typos) are included in red as suggested by the editor. We would like just to mention we added some more context on the applications of the IB in different domains and more experimentation on different datasets with different network architectures. Moreover, we found a way to exploit the value convergence phenomena in order to approximately achieve a desired compression level for both known and unknown IB curve shapes.

Sincerely,

Reviewer 3 Report

The paper should contain more elaborate introduction on application and role of the information bottleneck. The concept is well studied. Nevertheless, the context will improve readability. The proposed method should be compared against similar approaches (e.g. in a form of table) which could also help to depict more vividly of the research goal. Some more elaboration on an application of the proposed technique to more advanced Deep Learning architectures and datasets (larger than MNIST) could also help to grasp the contribution of the proposed approach.

Author Response

Dear Reviewer,

First of all, thank you very much for your comments and suggestions. We believe the manuscript has improved after we revised it with them in mind.

All the revisions of the manuscript (including typos) are included in red as suggested by the editor.  The additions based on your comments are:

We included more experiments with different datasets (including regression and non-deterministic classification tasks) with more complicated neural network architectures. We expanded the context of the information bottleneck including applications of the method in different domains and connections with other problems in the introduction. We could not include a table comparing our method to similar approaches. The reason for that is that, to date, the only method we know to explore the IB curve in scenarios where it could not be explored is the usage of the squared IB Lagrangian, which is a particular case of the convex IB Lagrangian, particularly of the power IB Lagrangian with alpha = 1.
We clarified that in the abstract changing 'other Lagrangians have been proposed' by 'another Lagrangian has been proposed', which could have been the source of the confusion.

Other than that, we found a way to exploit the value convergence phenomena in order to approximately achieve a desired compression level for both known and unknown IB curve shapes.

Sincerely,

Reviewer 4 Report

This manuscript investigates the Information Bottleneck problem.

First of all, one shall say that the Information Bottleneck problem aims to obtain relevant compressed representations of a random variable X, for predicting Y. This constrained optimization problem maximizes the information about the task, while guaranteeing a certain level of compression. The maximization of the Information Bottleneck Lagrangian allows solving the problem.

Therefore, it can be said that the theme upon which the research is conducted is current and interesting.

In this manuscript it is presented a general family of Lagrangian which allows for the evaluation of the Information Bottleneck curve for all scenarios. It is provided the exact one-to-one mapping between the Lagrange multiplier and the desired compression rate for known Information Bottleneck curve shapes, which allowed the elimination of the problem of solving the optimization problem for many values of the Lagrange multiplier.

The problem is well defined and methods are adequate and sufficiently explained. The research beneath the manuscript is substantial and replicable to a certain degree, given the fact that Authors made resources available.

The major accomplishment was the ability to solve the original constrained problem with a single optimization.

All things considered, I believe that this manuscript is a good scientific contribute and shall be published.

The document is readable, and globally well written, while some typos have been found.

Please consider substituting “we” for an impersonal subject.

Consider adding a comma “In this paper”, “In this section”, “Therefore, in representation-based machine learning methods”, “In all plots”

Consider changing “However, there is” to “However there is”

Consider correcting the typos “particluar”, “intuitevely”, “multipiers”, “conected”, “innacurate”, “perfomance”, “ressembles”, “unkown” and “Lagange”.

Consider substituting “need of multiple” to “need for multiple”

Consider to substitute “This corollary allow” by “This corollary allows”

Consider to change “Lagrangian allows to find” to “Lagrangian allows finding”

The word “difficult” is often overused. Consider changing to “challenging” (for instance).

Consider changing “In a deterministic or close to deterministic setting” to “In a deterministic or close to a deterministic setting”

Consider changing “an strictly” to “a strictly”

Author Response

(The authors gave the same response as above.)

Round 2

Reviewer 1 Report

The revision improves the experiment section quite a bit, by adding experiments on a few more datasets, and also addressing my question of proposing a method for targeting specific compression level, for both known and unknown IB curve, and discussing strong convexity. The paper as it is should be a good contribution to the IB community, and also facilitate practical application. I recommend acceptance.

The following are some minor points that may improve the paper.

Question:

In Theorem A1 (IB curve duality), the authors mentioned about the dual. Does there exist a simple relationship between (u, \beta) and (v, \beta_{dual}), such that they target the same point? It would be nice if the authors can derive such a relationship.

Some other related works:

I know of some related works that may be helpful to mention. It is up to the authors to determine whether to include.

In related works section:

IB for RL: Dynamics-Aware Unsupervised Skill Discovery, ICLR 2020.

IB for adversarial learning: Variational Discriminator Bottleneck: Improving Imitation Learning, Inverse RL, and GANs by Constraining Information Flow, ICLR 2019.

Also, in line 321, the authors note that the bottleneck variable performs an intrinsic clusterization in classification task, and the clusterization matches with the quantized performance levels observed. This is consistent with the theoretical study in a concurrent work "Phase Transitions for the Information Bottleneck in Representation Learning" in ICLR 2020.

Typo:

Line 276: Teorem -> Theorem

Author Response

Dear reviewer,

Thank you for your comments. We included the references you suggested in the appropriate places and wrote a simple relationship between $$\beta_u$$ and $$\beta_v$$ for fixed functions $$u$$ and $$v$$ in the section of 'Other Lagrangian families'. All the new revisions are in orange.

Sincerely,